# In-Depth Annotation of the *Drosophila Bithorax-Complex* Reveals the Presence of Several Alternative ORFs That Could Encode for Motif-Rich Peptides

**DOI:** 10.3390/cells10112983

**Published:** 2021-11-02

**Authors:** Magali Naville, Samir Merabet

**Affiliations:** IGFL, ENS-Lyon, 32/34 Av. Tony Garnier, 69007 Lyon, France; magali.naville@gmail.com

**Keywords:** lncRNA, smORF, altORF, ELM, SLiM

## Abstract

It is recognized that a large proportion of eukaryotic RNAs and proteins is not produced from conventional genes but from short and alternative (alt) open reading frames (ORFs) that are not captured by gene prediction programs. Here we present an in silico prediction of altORFs by applying several selecting filters based on evolutionary conservation and annotations of previously characterized altORF peptides. Our work was performed in the *Bithorax-complex* (*BX-C*), which was one of the first genomic regions described to contain long non-coding RNAs in *Drosophila*. We showed that several altORFs could be predicted from coding and non-coding sequences of *BX-C*. In addition, the selected altORFs encode for proteins that contain several interesting molecular features, such as the presence of transmembrane helices or a general propensity to be rich in short interaction motifs. Of particular interest, one altORF encodes for a protein that contains a peptide sequence found in specific isoforms of two *Drosophila* Hox proteins. Our work thus suggests that several altORF proteins could be produced from a particular genomic region known for its critical role during *Drosophila* embryonic development. The molecular signatures of these altORF proteins further suggests that several of them could make numerous protein–protein interactions and be of functional importance in vivo.

## 1. Introduction

Classical gene prediction programs are based on automated intron–exon annotations and comparison with cDNA sequences and/or genes from different organisms [1,2,3,4]. These computational methods led to the general finding that a surprisingly small fraction of the eucaryotic sequenced genomes (2/3% on average) corresponds to protein-encoding open reading frames (ORFs, [5]). The small number of these so-called “conventional” genes is in sharp contrast with the biological complexity of multicellular organisms [6]. Moreover, a large proportion of the genome was observed to be transcribed outside the conventional genes [5], suggesting that non-canonical coding sequences could escape detection from genome annotation pipelines. This non-conventional transcriptome, also assigned as non-coding (nc) RNAs, has for a long time been considered to be a transcriptional noise product [7]. ncRNAs are by now recognized as acting as various categories of RNA molecules (such as nucleolar-RNAs, enhancer-RNAs, circular-RNAs, micro-RNAs and long-ncRNAs) and to mediate fundamental cellular processes, including transcription and translation efficiency or mRNA stability [8,9,10,11,12,13]. Along the same line, mass spectrometry (MS)-based approaches systematically revealed that a significant proportion of the sequenced peptides did not match annotated proteins [14,15,16]. Most of these peptides originated from proteins that contained less than 100 residues (below the minimal length for a conventional protein) and could be produced from ORFs that do not contain an AUG-start codon or an optimal Kozak sequence [17,18,19]. These small ORFs (smORFs, also referred to as shortORFs, sORFs) are found in non-coding regions (ncRNAs and 5′ or 3′UTRs of mRNAs) and represent a large proportion of the transcriptome [20]. smORFs have also been called upstream ORFs (uORFs) when present in the 5′UTR region, with a general (but not exclusive, see also: [21,22]) role in the transcriptional and translational control of the associated downstream conventional ORF [23,24,25]. Bioinformatics analyses dedicated to smORFs in the plant *Arabidopsis* [26], yeast *Saccharomyces* [27] and fruit fly *Drosophila* [28] further predicted that smORFs could account for 3% to 5% of the protein-coding genes.

Because of their size, low abundance in the cell and more generally lack of tools (e.g., antibodies), a quite limited number of small peptides has been characterized at both the molecular and functional level in plant and animal species. A pioneer work showed that a small peptide of 71 residues, Polar granule component (Pgc), was critical to inhibit the somatic transcriptional program in *Drosophila* germ cells by forbidding the recruitment of a particular kinase that is necessary to activate the RNA polymerase II [29]. Other small peptides produced in *Drosophila* have been shown to regulate the activity of the transcription factor Shaven baby (Svb). These peptides, called Polished rice (Pri) or Tarsal-less (Tal), are synthesized from four altORFs of different sizes (ranging from 11 to 32 codons) and promote the cleavage of a N-terminal repressor domain in Svb. This removal allows Svb to activate the set of target genes involved in the formation of trichomes in the larval cuticle [30]. Pri/Tal peptides are also involved in the regulation of other developmental aspects in *Drosophila* [31]. Another alternative micropeptide produced from a lncRNA of the *Drosophila Bithorax-complex* (*BX-C*), called MSAmiP, has recently been described to be expressed in the male accessory glands, which produce a seminal fluid that is critical for controlling the postmating response of the female [32]. These examples illustrate the emerging and probably highly diverse molecular roles of non-conventional ORFs in vivo.

Accordingly, ribosome profiling analyses [33,34], together with the re-annotation of transcriptomes (based on algorithm for ORF prediction from transcriptions sequences) to re-analyze proteomic datasets [34,35], confirmed that several non-conventional mRNAs could lead to the production of proteins. For example, the re-annotation of proteomic datasets showed that as many as 174 771 non-conventional ORFs coding for 71 705 proteins could exist in *Drosophila* [35]. These observations definitively established the importance of considering non-conventional ORFs to better understand the basic products of the genome. The re-annotation analyses also revealed a novel category of non-conventional ORFs that has been called “alternative ORFs” (altORFs). In contrast to smORFs, altORFs encode for peptides that resemble more to canonical proteins in terms of genomic marks for synthesis (such as an ATG start codon and a Kozak sequence). They are also longer than 30 codons (and can be longer than 100 codons) and can be located on ncRNAs, in UTRs or in different reading frames from annotated CDSs in mRNAs [35,36].

Here, we describe a novel in silico prediction of altORFs that is specifically dedicated to *Drosophila BX-C*. Our extensive manual annotation was performed using stringent filtering criteria for sequence conservation and for genomic marks’ scores deduced from previously described altORF peptides (Aspden et al., 2014). Our goal was to propose a high-confidence list of altORFs, and to assess whether the selected altORFs could encode for potential functional proteins. To this end, we looked at several protein signatures such as the global structure or the presence of short interaction motifs.

Altogether, our work revealed that several alternative proteins of variable size, structure, and motif contents could be produced from different regions of *BX-C*. We propose that the critical role of *BX-C* during embryonic development could not only rely on the currently characterized lncRNAs and Hox genes but also on several and so far unexplored altORFs.

## 2. Materials and Methods

### 2.1. altORF Prediction

altORFs were predicted in the whole *BX-C* locus of *D. melanogaster* (chr3R:12,481,489-12,801,607 in dm3 assembly) and in orthologous regions of *D. simulans* (droSim1), *D. sechellia* (droSec1), *D. yakuba* (droYak2), *D. erecta* (droEre1), *D. ananassae* (droAna2), *D. pseudoobscura* (dp3), *D. persimilis* (droPer1), *D. mojavensis* (droMoj2), *D. virilis* (droVir2) and *D. grimshawi* (droGri1) identified with blastn. This region spans ca. 320 kb.

altORFs were identified on both strands using a home-made BioPython script requiring a minimal length of 30 amino acids. Briefly, this script searches for sequences longer than 90 nucleotides, beginning with an ATG and with no interruption by a stop codon. It sorts out the corresponding translated sequences. The full script can be provided upon request.

### 2.2. altORF Characterization

#### 2.2.1. Orthology and Age

Orthologous altORFs were identified between species by all-by-all peptides comparison using BlastP with default parameters and by requiring a maximum E-value of 1 × 10^−5^. Although a tBlastn search could have been more sensitive to detect orthologous sequences, it was not applied since it could not discriminate between altORF-containing and non-altORF-containing sequences. The minimal age of each ORF was deduced from the divergence time of the two most distant species in which it was found. Species divergence times were retrieved from [37].

#### 2.2.2. Nucleotidic Conservation

Conservation of the DNA sequences encoding the ORFs was computed from the PhastCons 15-way track available at UCSC, which evaluates evolutionary conservation in 12 *Drosophila* species, mosquito, honeybee and red floor beetle based on a phylogenetic hidden Markov model [38].

dN/dS ratios were computed for the final list of 48 altORFs using PAML [39]. Calculation was based on multiple alignments obtained with Muscle [40] and a phylogenetic tree reconstructed with PhyML [41] with the following parameters: Model-given amino acid equilibrium frequencies, no invariable sites, optimized across site rate variation, NNI tree searching, and BioNJ starting trees with optimized tree topology. Parameters for dN/dS calculation can be provided upon request (codeml.ctl). dN/dS must be taken with caution when dS < 0.01 (sequences too close, with not enough divergence) or dS > 2 (sequences too divergent).

#### 2.2.3. Transcription and Translation Features

Transcription promoters were predicted in the whole *D. melanogaster BX-C* region using the Neural Network Promoter Prediction program available on the Berkeley Drosophila Genome Project (BDGP) website (https://www.fruitfly.org/seq_tools/promoter.html, Reese 2001, last accessed on 14 October 2021), with default parameters. altORFs were attributed the closest upstream promoter, annotated by the intervening distance and its prediction score.

PolyA sites were searched in 150 nt downstream of the altORF in all the species were it was found using the PolyA_SVM program [42].

altORF expression was assessed in 8–10 h whole embryos by blasting them against RNA-seq data retrieved from modENCODE (SRX008010, SRX008249, SRX008252, SRX008273, and SRX008274 experiences). This stage was voluntary chosen to restrict the analysis and identification of altORFs that could potentially and specifically act with the Hox genes during their early patterning functions in the embryonic epidermis.

Kozak sequences were retrieved for each predicted altORF, and their relative strength was assigned according to [43].

#### 2.2.4. Overlap of altORFs with Other Genomic Features

Gene annotations (transcripts, exons, CDS) were retrieved from FlyBase (https://flybase.org/, accessed on 10 March 2015) and overlapped with DroMel altORF coordinates. DroMel altORFs were also overlapped with cDNAs retrieved from ModENCODE (http://www.modencode.org/, accessed on 5 August 2015), in sense or antisense orientation. altORF peptide sequences were finally compared to protein sequences Ubx, AbdA, or AbdB using BlastP to identify altORFs in frame with Hox coding sequences.

Transposable element annotations were retrieved from the Natural Transposable Element Project of the BDGP website (https://www.fruitfly.org/p_disrupt/TE.html, version 9.4.1. accessed on 17 September 2015), and overlapped with altORF coordinates.

Predicted altORFs were finally compared with different sets of peptides already published in the literature, including peptides from the PeptideAtlas database (http://www.peptideatlas.org/, last accessed on 14 October 2021), from Poly-ribo-seq approach [33], proteomic datasets [35], and smORFs [28].

### 2.3. Characterization of the Predicted Peptides

Predicted peptides were submitted to InterProScan (https://www.ebi.ac.uk/interpro/search/sequence/, last accessed on 3 October 2015) to search for putative conserved protein domains such as signal peptides or transmembrane helix. Potential signal peptides were predicted using the SignalP-5.0 Server [44] (http://www.cbs.dtu.dk/services/SignalP/, last accessed on 13 October 2015). Transmembrane helices were predicted using the TMHMM Server v. 2.0 [45] (http://www.cbs.dtu.dk/services/TMHMM/, last accessed on 13 October 2015).

Intrinsically disordered regions were predicted using UIPred2A server (https://iupred2a.elte.hu/plot_new, [46], last accessed on 11 May 2021).

Predicted altORF peptides were also compared to the full proteome by submitting them to a BlastP search against the nr database, with default parameters.

Eukaryotic Linear Motifs (ELMs) were identified using the ELM database (http://elm.eu.org/, last accessed on 7 June 2021) and FuzzPro (https://www.bioinformatics.nl/cgi-bin/emboss/fuzzpro, last accessed on 26 October 2015). The search was restricted to ELMs found in one of Ubx, AbdA, or AbdB proteins. All ELMs found in each altORF are given in Appendix A. Description of ELMs found in Ubx, AbdA, and/or AbdB is given in Appendix A.

A global “ELM score” was computed, for each ORF, as follows:(1)$$ELMscoreORF_{i}=\left(\underset{j}{\sum }\left(motifsj\mathrm{in}ORF_{i}/motifsj\mathrm{in}allORFs\right)/aminoacidlengthofORF_{i}\right)\times 10^{6}$$

This score attempts to quantify the mean occurrence of ELMs in each altORF by taking into account the length of the altORF as well as the frequency of each ELM in all the predicted altORFs.

## 3. Results and Discussion

### 3.1. Annotation of altORFs in BX-C

We have considered the 320 kb region of *BX-C* and applied a home-made BioPython script (see materials and methods) to search for all possible altORFs with two required criteria for altORFs: a minimal length of 30 codons and an ATG start codon [36]. In contrast to the previous annotations of altORFs, our prediction analysis was not restricted to ncRNAs and mRNAs (therefore considering the entire intergenic regions). This first step led to the prediction of 2086 altORFs (Figure 1A and Figure 2). We next considered altORFs that were not matching the conventional proteins Ultrabithorax (Ubx), Abdominal-A (AbdA) or Abdominal-B (AbdB) to discard altORFs that could correspond to any Hox isoform, achieving 2075 predicted altORFs in total (Figure 1A,B). In the second and third filtering steps, we arbitrarily decided that altORFs should be present in at least two different *Drosophila* species that are distanced by 10 M years or more during evolution (Figure 1A–C). These parameters allowed us to select evolutionary conserved altORFs, leading to a list of 1233 predicted altORFs (Figure 1A). To further increase the evolutionary confidence score, we analyzed the 1233 altORFs with PhastCons ([38] and materials and methods) and applied a threshold value based on the scores found with altORFs identified from previous genome annotation analyses [33,35] in our list (these scores are highlighted in red in Table 1). Several of these previously annotated altORFs have been confirmed by tandem mass spectrometry (MS/MS) in human [35] or *Drosophila* [33] cell lines, or by a sensitive ribosome footprinting approach called ‘Poly-ribo-seq’ in *Drosophila* S2 cells [33], underlining that their genome annotation scores could serve as a reference value for filtering altORF peptides with a higher synthesis probability. Of note, we could not use reference values from altORF peptides captured by proteomics approach and specifically located in *BX-C*: no MS/MS-captured altORF peptide is present in *BX-C* (https://openprot.org/p/browse, accessed on 1 October 2021) and the Poly-ribo-seq data could not be exploited in the context of *BX-C* since this genomic region is under strong transcriptional repressive state in S2 cells (these cells do not express any Hox gene; [47] and Appendix A). In conclusion, the PhastCons threshold value deduced from the previously annotated altORFs present in our list was 0.43 and this threshold was therefore applied to further filter our list, leading to 1150 predicted altORFs in *BX-C* (Figure 1A,B and Table 1).

Scores based on previously identified altORF peptides from genome annotation [33,35] present in the list and used for establishing the threshold values are highlighted in red.

The next filtering steps were based on genomic marks for transcription and translation. The selection filters were also deduced from the scores found with previously identified altORFs [33,35] present in our list. More precisely, we found that the promoter could not be located more than 278 nucleotides upstream of the start codon (ORF268 and ORFas200: Table 1) and we thus discarded all predicted altORFs that had a promoter located at higher distances from the first ATG. Along the same line, the minimum confidence score for the predicted promoter of previously identified altORFs of our list was 0.81 (Table 1), and we therefore did not consider altORFs that had a score below this value. Together, these two criteria (see also materials and methods) allowed us to restrict the list to 572 altORFs (Figure 1A,B). The next step consisted of applying a minimum score for the Kozak sequence (see materials and methods), which led to a list of 492 predicted altORFs (Figure 1A,B). Because the majority of the previously identified altORFs in this list did not show a relevant score for poly-adenylation sites (see materials and methods and Appendix A), we did not consider this parameter for a selection step. The list was then submitted to RNA-seq data scores, using reference thresholds found with previously identified altORFs: altORFs should have a minimum number of 5 reads over at least 40% of their length (Table 1, Figure 1A,B and Material and Methods). Here, we only considered RNA-seq data of stage 8–10 h embryos for considering the number of reads, which is a stage of high RNA expression and during which Hox genes are specifying the epidermis along the anterior-posterior axis. The rational was to identify altORFs that could potentially be involved in these early Hox patterning functions. The reference threshold for altORF covering by RNA-seq was deduced by considering both S2 cells and stage 8–10 h embryos. These last filtering steps led to a final list of 48 altORFs for which we also calculated the dN/dS ratio (i.e., the ratio of non-synonymous mutations on synonymous ones, a value smaller than 1 testifying for a negative selection pressure) to assess whether they were under purifying selection. The score indicated a negative selection (dN/dS < 1) for 74% of ORFs for which the calculation could be performed (with sequences that were not too similar (dS < 0.01) or too divergent (dS > 2): Appendix A and materials and methods).

Altogether, our filtering criteria eliminated more than 98% of the altORFs present in the initial list (2038/2086 altORFs have been discarded). The remaining 48 altORFs display conservation, transcriptional, and translational scores that are comparable with the scores found with previously predicted altORF peptides. We therefore considered the 48 altORFs with a high-confidence protein synthesis potential.

### 3.2. Structure/Motifs Predictions of Conserved Alternative Peptides of BX-C

The majority of the 48 selected altORFs are in *Ubx*, *abdA* or *AbdB* (for almost half of them), the rest being equally present in lncRNAs or intergenic sequences (Figure 2). They are also of highly variable size, from 30 to 274 codons, highlighting that they could encode for proteins with a comparable size to the conventional ones (Table 1).

To assess whether the predicted altORFs could encode for peptides with potential molecular functions, we looked at several protein signatures. First, we observed that there was not a strict rule regarding the global structure (see also material and methods). Although the majority (28/48) displays a global ordered structure, several altORFs are also predicted as globally disordered or with a mixture of ordered and disordered regions (Table 1 and Appendix A).

No conserved domains were found in the 48 altORF proteins (see material and methods), but 6 altORFs encode for a protein containing a predicted transmembrane helix (ORF268, ORFas452, ORFas561, ORFas566, ORFas293, ORFas1009: Table 1 and Appendix A). This protein feature has been shown to be enriched in altORFs [33]. It is interesting to note that one of these altORFs has been characterized in a pioneer analysis of *Ubx* genomic sequences [48]. This altORF, named ORF268, is located in the 5′UTR region of *Ubx* and encodes for a 71 aa-long peptide that has been predicted in several previous studies [33,35]. It also contains a signal peptide and a transmembrane helix (Figure 3). This altORF peptide therefore constitutes a very good candidate for future functional analyses in vivo. In particular, it would be interesting to know whether it could modulate the expression and/or function of *Ubx* given its genomic localization.

We next looked at the presence of Eukaryotic Linear Motifs (ELMs, also called short linear motifs, SLiMs), which have been classified in different types depending on their interaction properties and characterized functions [49]. Here, we focused on ELMs that were found in Ubx, AbdA and/or AbdB (Appendix A). We noticed that the large majority of previously captured altORF proteins of our list (11/15) had 4 or more ELMs of different types (Table 1 and Appendix A). The ORF44, which is in the first exon of the *AbdB-m* isoform (but in a different frame), is an extreme example with 125 ELMs belonging to 17 different types (Table 1 and Appendix A and Figure 4). These observations suggest that altORF-derived proteins have a general tendency to be rich in ELMs. Accordingly, we observed the same tendency with the other altORFs of our selected list (only 3 out of the 33 remaining altORF peptides have less than 2 ELMs: Table 1).

To better evaluate the number of ELMs independently of the protein size and the intrinsic complexity of each ELM, we established an ELM score that takes into account the number of ELMs present in the altORF compared to the total number of ELMs found in the initial 2086 predicted altORFs and reported to the size of the corresponding altORF protein (see materials and methods). This ELM score was weak (below 100) for few altORF proteins (8/48, two of which corresponding to previously predicted altORF proteins (ORF682 and ORF817): Table 1). A high (between 100 to 200: 19/48 altORFs) to very high (above 200: 21/48) score was found for the majority of our predicted altORFs. These scores underline that the number of ELMs is not directly linked to the length of the altORF protein and is therefore of potential significance. Interestingly, we found that a set of conventional ORFs with a range of sizes similar to our 48 selected altORFs displayed a comparable ELM score distribution (Appendix A). This observation underlines that a high ELM score is a hallmark of conventional proteins (as previously described: [50]) that is also found in our selected altORF peptides.

Several of these ELMs are largely distributed (like the USP7 binding motif or WW domain ligands motif; Appendix A), but others are more restricted, such as the SCF ubiquitin ligase binding phosphodegrons motif, which is found in AbdA and three altORFs in total (ORFs as430, 211, and 44; Appendix A), or the Cks1 ligand motif, which is found in AbdA and AbdB and five altORFs in total (ORFs as430, 211, 861, 389, and 44; Appendix A). Whether the common presence of more specific ELMs could be more significative of a common molecular function remains to be investigated. It is also interesting to note that two ELMs present in Ubx, AbdA and/or AbdB were never found in the 48 selected altORFs (the CtBP ligand motif and sumoylation site, Appendix A).

Finally, although we did not find any altORF with a significant score for conserved domains, one altORF, ORF682, contained a peptide sequence of 30 residues that is also found in the C-terminus of particular isoforms of Deformed (Dfd) and Sex combs reduced (Scr; Figure 5). The ORF682 is present in the coding sequence of *Ubx* (not in the same frame), has previously been predicted (Aspden et al., 2014), and encodes for a 60 aas long disordered protein that displays a reasonable ELM score (with two different ELMs outside the sequence found in Dfd and Scr: Figure 5 and Table 1). The presence of a common sequence between ORF682, Dfd and Scr is intriguing and suggests that it could mediate a common molecular function in the three proteins.

## 4. Conclusions

Our study revealed the presence of 48 altORFs with a high potential to encode for functional alternative proteins in *BX-C*. Probably many more non-conventional protein-encoding ORFs are present in *BX-C*. For example, our stringent selection criteria did not allow retrieving all previously predicted altORF proteins (Figure 1 and https://openprot.org/, last accessed on 18 October 2021), highlighting that alternative proteins could be less conserved within the *Drosophila* evolutionary tree and/or produced using more flexible genomic annotation marks. Along the same line, we did not consider smORFs. For example, 5098 smORFs with a minimal size of 10 codons could be predicted in *BX-C*. Several of those predicted smORFs could likely encode for small peptides, as described for MSAmiP [32].

Several of our 48 predicted altORF proteins contain molecular features that are highly suggestive of putative functions in vivo. These include the presence of a transmembrane helix or a general propensity to contain several ELMs. However, the molecular versatility of ELMs [51] does not allow the prediction of the precise molecular function of the altORFs, or whether a functional link could exist with Hox or any other specific protein in vivo. In any case, our work confirms the extraordinary potential for the genome to produce several alternative proteins. By restricting to a specific genomic region, it also identifies high-confidence altORFs (such as the ORF268) for future molecular and functional analyses in *Drosophila*.

## Figures and Tables

**Figure 1 cells-10-02983-f001:**
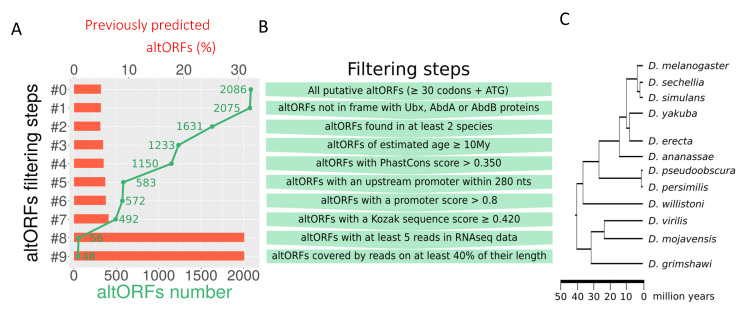
Identification of altORFs in the *BX-C* region of *Drosophila melanogaster*. (**A**). Total number of predicted altORFs (green curve) across the 9 different filtering steps. The corresponding percentage of previously predicted altORF peptides [33,35] is indicated (red bars). (**B**). Criteria applied across the different filtering steps. The threshold values used for selecting most relevant altORFs were based on scores deduced from previously predicted altORF peptides (see also materials and methods). (**C**). Phylogeny of the 12 *Drosophila* species used for conservation analyses (adapted from [37]).

**Figure 2 cells-10-02983-f002:**
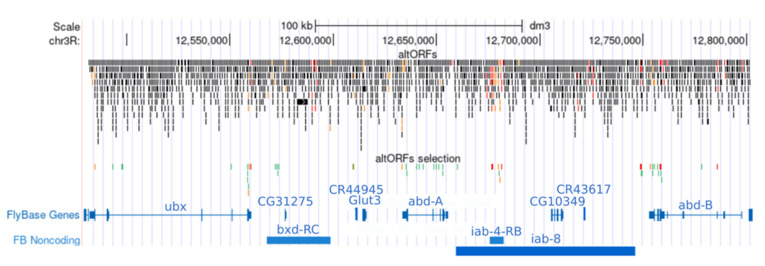
Distribution of the predicted altORFs in the *BX-C* region. Each altORF is represented as a small bar. The initial (2086, upper lines) and final (48, lower lines) number of predicted altORFs is shown. altORFs that have previously been identified are highlighted in red [33] or orange [35]. lncRNAs (blue bars) and CDS (blue lines with UTRs and exons) are indicated.

**Figure 3 cells-10-02983-f003:**
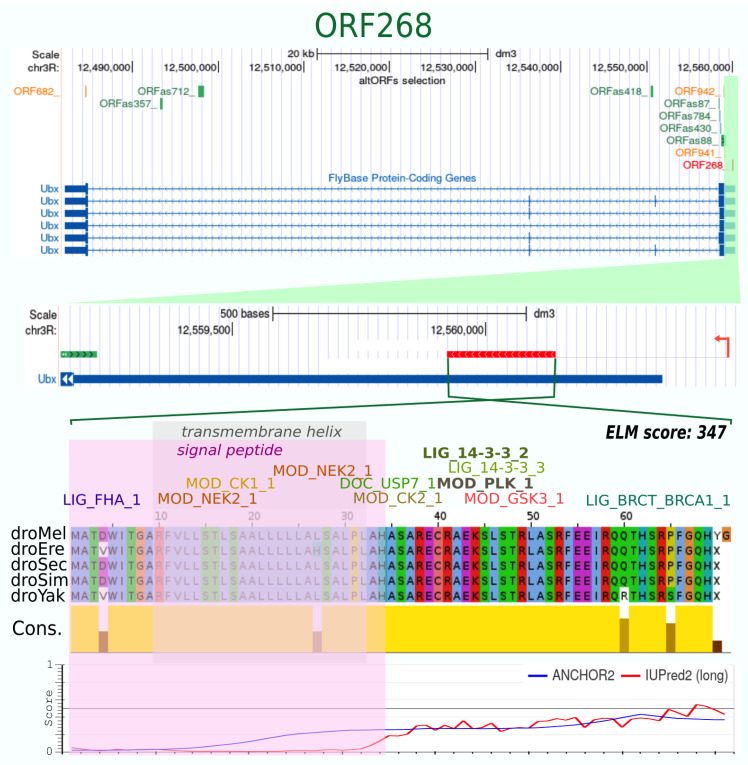
Characterization of the ORF268. This altORF is in the 5′UTR of *Ubx*, in the same orientation. The putative promoter is indicated (red arrow). It is found in 5 species and is highly conserved (yellow graph below the protein sequence alignment). The 71 aas long peptide is globally ordered. It contains a signal peptide and a transmembrane helix (pink and gray shadows, respectively), as well as several ELMs of different types. This altORF has previously been described in a pioneer study of the *Ubx* genomic region [48].

**Figure 4 cells-10-02983-f004:**
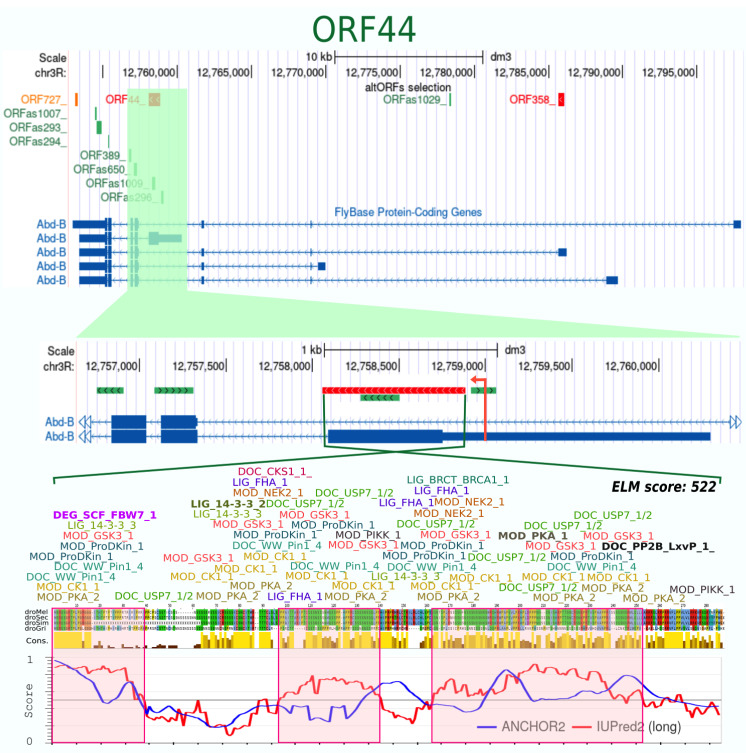
Characterization of the ORF44. This altORF overlaps with the 5′UTR and first coding exon of one *AbdB* transcript. It is in the same orientation but in a different reading frame. This altORF was found in 4 *Drosophila* species. The 274 aas long predicted protein displays three disordered regions (highlighted in pink) and numerous ELMs of different types.

**Figure 5 cells-10-02983-f005:**
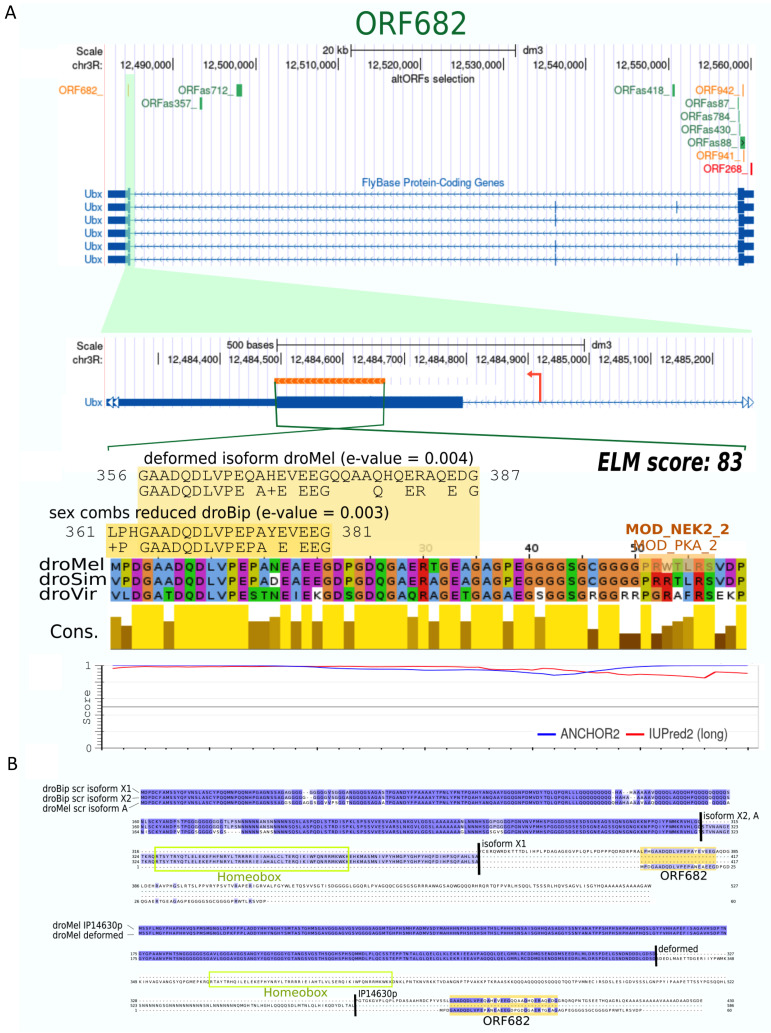
Characterization of the ORF682. (**A**). This altORF overlaps with the *Ubx* last coding exon and the beginning of its 3′UTR, in the same orientation but in a different reading frame. ORF682 is found in 3 *Drosophila* species. The 60 aas long predicted protein is globally disordered and contains only two ELMs of different type. (**B**). A peptide sequence present in the ORF682 is found in an isoform of *D. bipectinata Sex comb reduced* (*Scr*; isoform *X1*) and *D. melanogaster Deformed* (*Dfd*; isoform *IP14630p*). These two isoforms result from a frameshift of the last part of the gene encoding the homeobox domain (black bars).

**Table 1 cells-10-02983-t001:** List of the 48 selected altORFs following the different filtering steps based on conservation, transcription, and translational marks criteria.

Name	Distribution and Conservation					Transcription and Translation Features					Peptide Characterization							
	Length(aa)	Strand	Age(My)	Droso	PhastCons_Score	PromScore	PromDist	KozakScore	Reads	%Reads	Aspden et al.	Samandi et al.	TotalELMs	DifELMs	ELMScore	Signal Peptide, TMhelix	IUPred_structure	BlastP_nr
ORF682_	60	+	40	3	0.954	0.96	−246	0.643	59	55		yes	2	2	83		disordered	Scr, Dfd
ORFas357_	108	-	27	8	0.776	0.85	−191	0.874	8	46			14	8	144		ordered	
ORFas712_	215	-	14	5	0.718	0.98	−23	0.538	6	45			32	13	184		ordered	
ORFas418_	88	-	10	5	0.459	0.96	−143	0.448	5	58			13	8	144		18% disordered	
ORFas87_	46	-	27	5	0.623	0.81	−47	0.755	97	54			5	4	188		ordered	
ORFas784_	49	-	14	6	0.920	0.93	−84	0.776	53	76			4	4	220		ordered	
ORFas430_	51	-	10	4	0.921	0.93	−16	0.916	90	82			16	10	936		unclear	
ORFas88_	194	-	10	5	0.958	1.00	−232	0.881	193	60			18	12	410		ordered	
ORF942_	38	+	40	6	0.986	0.81	−48	0.434	77	98		yes	5	4	195		disordered	
ORF941_	37	+	10	5	0.984	0.81	−171	0.490	55	97		yes	2	2	228		disordered	
ORF268_	71	+	10	5	0.755	0.96	−278	0.594	123	88	yes	yes	15	10	347	1 Signal peptide, 1 TMhelix	ordered	5′-Ubx
ORFas449_	61	-	40	5	0.699	0.90	−121	0.944	9	77			8	6	234		disordered	
ORFas450_	36	-	40	10	0.995	0.99	−9	0.986	6	51			1	1	13		32 % disordered	
ORFas803_	116	-	10	4	0.544	0.92	−79	0.503	10	64			14	9	167		ordered	
ORFas452_	75	-	10	3	0.462	0.92	−167	0.825	8	60			15	9	196	1 TMhelix	7 % disordered	
ORF211_	48	+	10	3	0.659	0.84	−8	0.643	6	53			10	9	989		36 % disordered	
ORFas137_	113	-	10	5	0.616	0.94	−72	0.699	9	65		yes	22	10	197		ordered	
ORF861_	41	+	10	5	0.765	0.88	−202	0.427	14	89		yes	8	8	432		ordered	
ORFas875_	38	-	10	5	0.906	0.95	−165	0.434	185	98			1	1	38		ordered	
ORFas165_	62	-	14	6	0.936	0.93	−141	0.727	82	53			6	3	74		ordered	
ORF172_	80	+	10	5	0.607	0.97	−85	0.790	12	81			10	8	233		ordered	
ORF497_	109	+	40	9	0.480	0.91	−189	0.839	7	68			17	9	233		ordered	
ORFas170_	68	-	10	4	0.511	1.00	−96	0.434	7	91			12	10	302		ordered	
ORF847_	70	+	10	4	0.490	0.88	−130	0.427	5	65			28	10	461		14 % disordered	
ORFas175_	114	-	40	11	0.628	0.81	−177	0.503	16	64			14	9	132		ordered	
ORF842_	34	+	10	5	0.815	0.89	−118	0.762	5	79			11	8	465		unclear	
ORFas894_	152	-	40	7	0.617	0.93	−42	0.497	93	59			13	9	180		6 % disordered	
ORFas538_	57	-	10	5	0.974	0.93	−97	0.650	79	94			5	5	1177		unclear	
ORFas545_	47	-	10	5	0.600	0.84	−36	0.727	7	42			4	4	247		ordered	
ORFas561_	72	-	10	3	0.647	0.82	−112	0.455	8	58	yes	yes	8	6	168	1 TMhelix	25 % disordered	
ORFas200_	54	-	10	5	0.484	0.81	−278	0.441	6	100	yes	yes	6	5	141		ordered	
ORFas566_	80	-	10	5	0.682	0.98	−265	0.734	13	90		yes	17	11	409	1 TMhelix	ordered	
ORF459_	46	+	10	4	0.571	0.85	−205	0.713	6	100			3	3	131		48 % disordered	
ORFas567_	32	-	10	5	0.582	0.88	−47	0.580	8	86		yes	7	7	235		ordered	
ORF817_	49	+	10	4	0.595	0.84	−251	0.657	8	50	yes	yes	3	3	66		37 % disordered	
ORF52_	242	+	10	5	0.431	0.98	−93	0.629	51	50	yes	yes	30	12	120		5 % disordered	
ORF393_	241	+	10	5	0.442	0.89	−159	0.699	27	63			30	13	153		ordered	
ORF727_	53	+	10	4	0.539	0.93	−210	0.552	11	96		yes	5	4	151		disordered	
ORFas1007_	30	-	10	5	0.935	0.98	−105	0.832	39	93			5	5	287		ordered	
ORFas293_	100	-	10	4	0.523	0.95	−134	0.427	187	84			9	5	65	1 TMhelix	ordered	
ORFas294_	34	-	10	2	0.691	0.95	−112	0.441	69	66			4	4	161		ordered	
ORF389_	51	+	14	3	0.499	0.99	−129	0.755	40	46			12	8	352		67 % disordered	
ORFas650_	75	-	10	5	0.839	1.00	−234	0.538	118	54			6	5	965		ordered	
ORF44_	274	+	40	4	0.906	0.81	−100	0.818	227	45	yes		125	17	522		61 % disordered	
ORFas1009_	75	-	10	4	0.938	0.98	−181	0.762	79	47			0	0	0	1 TMhelix	ordered	
ORFas296_	49	-	10	5	0.697	0.84	−1	0.853	17	96			7	6	135		ordered	
ORFas1029_	39	-	14	5	0.632	0.90	−9	0.965	6	58			1	1	31		ordered	
ORF358_	133	+	40	8	0.484	0.86	−160	0.748	43	47	yes		10	9	100		ordered	

ORFs are listed in the order of their cytological location in *BX-C* (see also Appendix A). Information given in the different columns are categorized in three classes (see also materials and methods): 1—Distribution and conservation (highlighted in light orange): - Length (in amino acids); - Strand: relative to the orientation of *Ubx*, *abdA*, and *AbdB*, which are on the reverse strand; - Age: deduced from the divergence time of the two most distant species in which the altORF was found. Species divergence times were retrieved from [37]; - Number of *Drosophila* species where the altORF can be found (see also Appendix A); - PhastCons score; 2—Transcription and translation features (highlighted in medium orange): - Score of the closest upstream Promoter; - Distance of the closest upstream Promoter; - Score of the Kozak sequence; - Number of RNA-seq reads; - % of the altORF covered by RNA-seq data reads; 3—Peptide characterization (highlighted in dark orange): - The corresponding altORFs has previously been predicted (‘yes’) from [34]; - The corresponding altORFs has previously been predicted (‘yes’) from [20]; - Total number of Eukaryotic Linear Motifs (ELMs) found in the encoded alternative protein; - Total number of different types of ELMs found in the encoded alternative protein; - ELM score; - Presence of a signal peptide and/or a transmembrane (TM) helix in the encoded alternative protein; - Global prediction of ordered and disordered regions deduced from IUPred; - Peptide sequences found by BlastP in other proteins. One sequence from the altORF682 was found in the Hox proteins Deformed (Dfd) and Sex combs reduced (Scr). The altORF268 has previously been annotated in the promoter region of *Ubx* [48].

## Data Availability

Predicted ORFs from [35] are available on https://www.roucoulab.com/p/downloads (accessed on 1 October 2021) with no restrictions. Predicted ORFs from [33] are publicly available in the Gene Expression Omnibus (GEO) database under accession GSE60384.

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
