# Peer review of "In-Depth Annotation of the Drosophila Bithorax-Complex Reveals the Presence of Several Alternative ORFs That Could Encode for Motif-Rich Peptides"

_cells, 2021, doi:10.3390/cells10112983_

Round 1

Reviewer 1 Report

The manuscript by Naville & Merabet reports a bioinformatic study exploring the coding potential of the Bithorax-complex (BX-C) in flies. BX-C is one of the two HOX cluster in Drosophila and it plays key roles in the development and differentiation of posterior segments. It represents a large genomic region (320kb) that encodes three HOX proteins (Ubx, Abd-A and Abd-B), as well as several long non coding RNAs and other poorly characterized transcripts. There is a growing body of evidence to show that the actual coding potential of cellular RNAs has been underestimated. A substantial proportion of alleged non coding (nc) RNAs produce peptides translated from small open reading frames (smORFs), which can fulfil important functions. Recent work has well demonstrated the role of smORF peptides in the control of development, including for a transcript from the Drosophila BX-C. In addition, many studies have established that even bona fide mRNAs can harbor supplementary/alternative ORFs, located in the 5’UTR or 3’UTR of the main ORF, as well as within introns, or exons from a different frame.

Merabet is well recognized for his work on HOX proteins and the current study aims at exploring the existence of putative alternative ORFs (alt-ORFs) within the BX-C genomic region. The authors analyzed a series of available data (expression, sequence conservation, proteomics, Ribo-seq, etc..) using a multistep pipeline, with filtering criteria inspired by the current knowledge on alt-ORFs. These analyses predict a set of 48 putative alt-ORFs. Interestingly, one of these candidates (#268, 71aa) that is located in the 5’UTR of Ubx had also been identified by previous studies providing experimental evidence for its translation. Although none of 48 alt-ORFs match known functional protein domain, some of them might include transmembrane helixes. The authors also found that alt-ORFs often comprise sequences that match small protein interaction motifs (eukaryotic linear motifs ELMs, also called short linear motifs, SLiMs). Puzzlingly, they also found one altORF (#682, 60 aa) of Ubx that matches peculiar isoforms of Sex comb reduced (Src) and Deformed (Dfd), two genes of the other HOX cluster ANT-C.  

In sum, the paper reports a smart series of complementary approaches to predict alt-ORFs from the Bithorax complex, and bioinformatics seems properly done. Besides the lack of functional characterization, which limits the scope of these studies, here are some additional suggestions to improve the manuscript.

  • One of the criteria used to filter alt-ORF candidates relates to promoter distance (set to 280nt, Fig 1B). The relevance of this parameter (that seems to have had a strong impact on the number of selected candidates) and how the threshold has been defined should be better explained. In addition, this criterion seems unlikely to apply to ORF682, located in the last coding exon of Ubx, i.e., more than 75kb downstream of the promoter!
  • It also remains unclear why the authors focused on a single embryonic stage (8-10h) to interrogate RNA-seq data, given the broad expression of Ubx, AbdA and AbdB throughout Drosophila development and adulthood (and the availability of corresponding data).
  • The pipeline used to assay for evolutionary conservation should be clarified. Since alt-ORFs are expected to be underestimated, search using BlastP might be not optimal. For example, the authors can use tBlastn to compare the 48 candidates in D. melanogaster to nucleotide databases in other species.

In addition, nucleotide alignments provided by PhastCons tracks of UCSC are often of weaker quality when considering more distant species. Therefore, the reported lack of conservation might be, at least in part, due to alignment issues. It would be of interest to analyze in more details at least some of the candidates, using optimized alignment, in order to strengthen the conclusion on both evolutionary conservation and direction of natural selection.

  • Related to the latter point, a dN/dS ratio <1 infers that non-synonymous mutations are likely under negative selection; in other words, that there is a selective pressure to maintain the amino acid sequence. The corresponding sentences (P4) should be rephrased to clarify this point, in particular for non-specialists. Also, the standard dN/dS test has some drawbacks, notably for short sequences. More recent methods (e.g. PhyloCSF, PMID: 21685081) have been shown to perform better for detecting purifying selection on small protein-coding sequences and they should be considered for reanalyzing best candidates.

Minor points

  • P2. that "MSAmiP, has recently been described in the male seminal fluid" is inaccurate. MSAmiP is expressed in the accessory glands that produce the seminal fluid, but there is no experimental evidence for the presence of MSAmiP in the fluid (PMID: 33876742).
  • P7. “Statistical significance” is vague and likely inappropriate. If a statistical test has been performed, the authors should state it and instead provide the p value. In any case, the term “enriched” seems adequate to the findings. An additional control would be to evaluate this ELM score in a random set of bona fide proteins (ideally of similar size repartition), as means to assay whether the high score found in 48 alt-ORF candidates was representative of known coding sequences, or might be a specific feature of alt-ORF proteins.
  • Genome browser snapshots of Figure 3-5 should include a scale for the whole locus and zoom on the transcript harboring alt-ORF.
  • Highlighting location of the promoter in Figures will also be helpful. 

Reviewer 2 Report

The manuscript entitled « The Drosophila Bithorax-Complex contains several alternative ORFs that encode for motifs-rich-peptides » by M. Naville and S. Merabet aims to describe and identify unannnotated (s)ORF peptides potentially produced by the BX-C region.
The paper is in general well written and well presented. I have no criticism regarding this. The data presented are of interest but only predictive. 
My major concern is their adequation with the publication policy of « Cells » (and the chapter in which it is suspected to be published) to bring enough novel data to justify publication. In fact, the manuscript presents only predictions. There is no functional evidences or analyses which constitute the real force of Drosophila model system, ie presenting experimental evidences of altORF function/expression of either by gain or loss of function experiments. Presenting experimental data demonstrating the translability of these ORF is, from my point of view, necessary.

Specific points :
- Change the title for a more appropriate one : « the Drosophila…. ORFs potentially encode motifs-rich-peptides »

- Abstract lane 11 :
Our work suggest that a number of altORF proteins might be produced…

- Introduction lane 20 : ..proteins that contained..

- Introduction : 3rd chapter : the studies on pgc performed by Nakamura et al must be cited (PMID: 18200011; PMID: 8953037). They also constitute important pionering studies on short ORF coding capacities.

- the authors cite the study  by Saari and Bienz (1987) as an experimental evidence of the presence of the peptide268.  However, the study map the TSS of UBX mRNA. They identifies the sORF leader sequence. The authors further said : « It is possible that this open reading frame in the Ubx mRNA leader serves some regulatory function, and the question arises whether it does so by acting in cis or in trans. The high sequence conservation observed in the 5' part of the D. funebris Ubx leader ,(near-identity up to position +86; Wilde and Akam, 1987) indicates functional significance of this region. The conservation however includes only the first 23 amino acids of the putative leader peptide; termination of the D. funebris open reading frame occurs after 30 amino acid codons. This may suggest that it is not the Ubx leader peptide per se that provides a potential regulatory function. On the other hand, initiation at the leader peptide AUG codon could affect Ubx protein expression in a similar way as has been observed for GCN4 expression in yeast (Muller and Hinnebusch, 1986) or recently for SV40 early protein expression (Khalili et al., 1987). Translation initiation at the upstream leader AUG codons in these cases inhibits efficient expression of the cis-linked protein downstream. » …. Which is far from the proof for a the translation of the peptide. What they say is still completely true and under debate since the function of uORF are principally considered as cis regulating elements. Translation and function of peptides resulting from uORF translation are still debated.

- As far as I know, the data from Samandi et al 2017 are only altORFs predictions in Drosophila… There is no actual Mass Spectrometry data produced from this study which could provide any identification of peptides produced from alternative ORFs in Drosophila. Correct the ms accordingly especially p5  (bottom) where the authors wrote « annotated from proteomic datasets (Samandi et al., 2017). ». In its present form, the sentence is not correct.

- If these data exist, the authors must show the MS and rib-seq profiles and not only an indication in table 1 ; There is no reference from where they got them and I could not find from where they got this info. (data presented in samandi are in principle released in openprot database; and there is no MS peptide described (see attached screen shot)) They should provide as sup data the translation of the peptides identified in Aspden et al.

Round 2

Reviewer 2 Report

> Unfortunately, the authors corrected only few of my requests. And not the most important ones. 
It is still not clear from where they got drosophila MS datasets. If any? This is extremely important as they claim that some of the predicted altprots are detected by MS experiments.
Among the potential altprot identified in Bx region, the authors listed as "MS detected peptides" (table 1 and fig2) the peptides identified in Samandi et al 2017. However, these peptides were identifed in this paper only from predictions... In Aspden et al,  only the files from rib-seq experiments are available. there is no files or databases referenced regarding MS experiments.
In Samandi et al 2017, it is true that there is some MS data but for the HUMAN genome. In drosophila, the altprot are only from predictions. this must be clearly indicated;
The authors must be more precocious and change the text accordingly:
- by removing the column MS in table 1 (or providing the correct reference or web site from where MS data can be downloaded). they can either cite samandi et al of change the head of the column for "predicted" or "re annotated" but certainly not "MS".
- changing p5 : ..."and applied a threshold value based on the scores found with previously captured altORFs by Poly-Ribo-Seq or re-annotation of MS datasets (Aspden et al., 2014; Samandi et al., 2017)".... to "and applied a threshold value based on the scores found with previously captured altORFs by Poly-Ribo-Seq or genome re-annotation (Aspden et al., 2014; Samandi et al., 2017)
- legend of fig2: change "...that were also previously identified by Poly-Ribo-Seq (Aspden et al., 2014) or annotation of mass spectrometry datasets (Samandi et al., 2017)" to "that were also previously identified by Poly-Ribo-Seq (Aspden et al., 2014) or re-annotation (Samandi et al., 2017)"

> the ref (PMID: 18200011) for the first description of the pgc sORF peptide is still not cited.

Author Response

We have corrected our manuscript (highlighted in yellow in the text) and the Table 1 according to the reviewer’s recommendations. We agree with the reviewer that the altORF peptides identified in Samandi et al. correspond to prediction only in the context of Drosophila, and not to the re-annotation of captured peptides from MS data, as it is for human altORF peptides. In fact, this point led us to also reconsider our definition of “captured peptides” from the Poly-ribo-seq approach of Aspden et al. There was indeed also a confusion between “captured” and “predicted”, and the notion of “previously captured” does not apply anymore. This is now more clearly stated in the main text (with a dedicated new paragraph to better explain our filtering strategy and a new Supplementary Figure 1) and the legend of Table 1. We sincerely thank the reviewer for his/her expertise and vigilance.

We also included the missing reference, together with a brief description in the third part of the introduction, about the pioneer described role of the Pgc small peptide in Drosophila germ cells.

Round 3

Reviewer 2 Report

the ms is now scientifically exact and can be accepted for publication if it fits with the journal criteria.